# First Molars–Incisors Rate and Pattern of Bone Loss: A Cross-Sectional Analysis of CBCT Images

**DOI:** 10.3390/diagnostics12071536

**Published:** 2022-06-24

**Authors:** Faraedon Mostafa Zardawi

**Affiliations:** 1Department of Periodontics, College of Dentistry, University of Sulaimani—Old Campus, Madam Mitterrand Street, Sulaymaniyah P.O. Box 1124-30, Kurdistan Region, Iraq; faraedon.mostafa@univsul.edu.iq; Tel.: +964-770-226-3062; 2Faculty of Dentistry, Qaiwan International University, Sulaimani P.O. Box 1124-30, Kurdistan Region, Iraq

**Keywords:** bone loss, CBCT, first molars, incisors, periodontitis

## Abstract

Background: Periodontitis causes attachment and alveolar bone loss; hence, this study aimed to determine the prevalence, frequency, and pattern of bone loss at first molar–incisor areas using Cone Beam Computed Tomography (CBCT) images. Methods: A retrospective, cross-sectional analysis was conducted of 250 randomly selected CBCT images of clearly defined full arches of patients aged from 18 to 70 years who were divided into six age groups and into male and female groups. Four sites around each tooth were scanned at several accesses for bone loss detection. Distance beyond 2 mm apical to the cementoenamel junction to the level of the remaining bone was considered to indicate bone loss. The Shapiro–Wilk test was used to test the normality of the data, and statistical tests were applied for data analysis at the 0.05 *p*-value level. Results: The rate and amount of bone loss within the examined sample were relatively high. The examined images generally revealed a higher rate of bone loss on proximal than on labial/buccal and lingual/palatal surfaces of the first upper and lower molars. The highest amount of bone loss among all the teeth scanned in this study was seen on the mesial and distal bone of mandibular incisors, 4.36 mm and 4.31 mm, respectively, exceeding that in the labial and lingual bone, 3.23 mm and 1.89 mm, respectively, and it was highly horizontal rather than vertical in pattern. Conclusions: Based on 250 randomly selected CBCT images of clearly defined, full upper and lower arches scanned for this study, it was concluded that the rate and amount of horizontal bone loss were less than vertical bone loss and was focused mainly in the interproximal areas of the first molars. However, the highest recorded amount of bone loss was at the proximal and labial aspects of the mandibular incisors. Furthermore, younger age groups displayed significantly higher rates and amounts of bone loss than older groups, with a slight predilection for males.

## 1. Introduction

The dysbiotic alteration of the microbial community–host symbiosis at the dentogingival area results in periodontitis, which leads to attachment and alveolar bone loss. Recently, periodontitis was categorized under several common forms [1,2], among which was the Molar/Incisor Pattern (MIP) [2]. This form of periodontitis is rare in prevalence and presents early in life and is associated with minimum plaque accumulation and, infrequently, with little gingival inflammation and severe, rapid periodontal destruction [3]. The high manifestation of MIP in members of the same family is determined by a genetic characteristic [4].

Appropriate assessment of the amount and morphology of the remaining bone is basically achieved by probing around the teeth and radiographic imaging. These two diagnostic tools have some potential drawbacks in the determination of the amount and morphology of the remaining bone [5,6,7], the major limitation being the inability of these traditional methods of examination to detect three-dimensionally the complex architecture of the area, which consequently makes it hard to justify the presence or absence of some osseous defects. Meanwhile, CBCT is a high-quality Three-Dimensional (3D) imaging system that is able to capture detailed images of dental and periodontal structures and overcome the limitations of 2D radiography. Pattern bone loss was categorized into horizontal, vertical, and furcation. Further, crater, dehiscence, and fenestration were also added to this category. Angular bone loss was also classified according to the number of remaining walls into one, two, and three walls and combined when the number of defected walls was greater at the apex [8]. Detection of osseous defects using 2D radiography is challenging due to the limitation of examination tools to exhibit the 3D architecture of the dentoalveolar area.

A distance of 2 mm apical to the cementoenamel junction is suggested to be the normal level of the crest of the interseptal bone [9,10,11]. This distance is dedicated for supra crestal tissue attachment (biologic width), and an apical distance beyond 2 mm indicates the presence of interseptal bone loss. Several comparative studies have justified the advantages of CBCT over 2D radiography in determining the amount and pattern of osseous defects [12]. For example, Pour et al. (2015) recognized that CBCT enables precise measurement of bone alveolar loss comparable to surgical findings, which can be applied for the diagnosis of osseous defects in periodontitis as an adjunct to the clinical assessment [13].

A retrospective panoramic study for alveolar bone loss among young adults in Sulaimani City, Iraq, was conducted by Zardawi et al. (2014) based on 1072 panoramic images retrieved from the archive of the College of Dentistry, Department of Radiology. The rate of bone loss was significantly high, with 30.2% of the total images showing bone loss at 1 site or more, 14.6% of the images revealing bone loss at 1–3 sites, and 15.6% demonstrating bone loss at more than 3 sites [14]. The prevalence of bone loss was not trivial, but significant in this city. Therefore, we believed that it was necessary to conduct a CBCT survey on a randomly selected sample of CBCT images for the purpose of matching the outcome of the current study with the previous results. So far, no studies have interpreted the MIP of bone loss using CBCT. To the best of our knowledge, this is the first study conducted with the aim of determining the prevalence, frequency, and pattern of bone loss at first molar–incisor areas using CBCT.

## 2. Materials and Methods

### 2.1. CBCT Imaging

Images for 250 subjects, namely patients whose ages ranged between (18 and 70). years, were randomly obtained from the radiographic archive of a private dental center. The CBCT images were captured by Sirona GALILEOS comfort—2016 and set at 98 Kv, 25 mAs, Field of view GALILEOS Compact (12 × 15 × 15) cm^3^ with 3D Resolution (isotropic voxel size) 0.3 mm.

### 2.2. Study Design and Measurements

In a retrospective, cross-sectional analysis, alveolar bone loss was inspected at four aspects (mesial, distal, mid-buccal, mid-lingual) around all the remaining teeth, highlighted at the molar/incisor regions, and compared to the presence of sites around the other teeth. CBCT scanning was performed in the following views: cross-sectional for the detection of osseous defects (buccal and lingual), axial access for furcation detection, and tangential view for inspecting and measuring interproximal osseous defects (Figure 1). The amount of the bone loss was measured digitally from 2 mm apical to the Cementoenamel Junction (CEJ) to the level of the remaining bone using the software provided by Sirona GALILEOS comfort—2016. Osseous defects were identified as vertical or horizontal, while furcation defects were simply recorded as the presence or absence of bone loss (orthopantomography images were used for the detection of bone loss at the furcation areas; further confirmation was performed by CBCT) [15].

### 2.3. Inclusion and Exclusion Criteria

Inclusion criteria included undistorted, clear CBCT images of full arches for both upper and lower jaws. Unclear blurred images that did not identify CEJ clearly or were distorted or overlapping were excluded. In addition, images of 3rd molars and those captured for local sections and for patients less than 18 years old were excluded from the present study.

### 2.4. Study Registration

The study proposal was registered with the Scientific Committee of the College of Dentistry, and ethical approval was obtained from the Ethical Committee of the College of Dentistry (N10 at 1 September 2021).

### 2.5. Inter- and Intra-Examiner Calibration

Inter-examiner calibration was based on a training course on the interpretation of CBCT images provided by the radiologist at the Radiology Department of the same center that provided the CBCT images, and inter-examiner calibration was achieved at a level of 85% competency after two weeks of this training. In addition, intra-examiner calibration for the interpretation of the CBCT images was achieved at a level of 87% competency after a duration of one week.

### 2.6. Sample Size Calculation

Since no similar study was performed previously, sample size calculation was performed using the G*Power 3.1 program, at a *p*-value 0.05, a power of 90%, and an effect size of 0.2; the sample size estimated by the G*Power software was 216 CBCT images, and to increase the level of confidence for the statistical data, 250 CBCT images were used in this cross-sectional study.

### 2.7. Statistical Analysis

The Shapiro–Wilk test was used to test the normality of the data. The Mann–Whitney 2-tailed test was used to determine the level of significance between females and males and Friedman’s test for furcation analysis. The Kruskal–Wallis test was used to compare the mean values of bone loss and pattern (horizontal and vertical). The ANOVA test was used to determine the statistical differences between age groups. The results of these tests were considered significant at a *p*-value level of 0.05.

## 3. Results

The demographic profile of the study sample is presented in Table 1 below. The study sample included males (*n* = 97) and females (*n* = 153), whose ages ranged from (18 to 70) years, with a mean age of (40.68 ± 13.12). The CBCTs were divided into six age groups. In this study, the alveolar bone around a total of (*n* = 2578) teeth and (*n* = 11,312) sites was examined to detect any kind of bone loss at the four aspects, including (*n* = 668) first molars in total that comprised (*n* = 370) and (*n* = 298) maxillary and mandibular first molars, respectively. A total of (*n* = 1910) incisors were examined, comprising (*n* = 943) and (*n* = 967) maxillary and mandibular incisors, respectively (Table 1). Thorough interpretation of (*n* = 10,312) sites on these images produced the following findings. The rate and amount of bone loss within the examined sample were relatively high, with 23.2% of the total teeth examined in this study: 58 (1%) first molars, 540 (8.9%) incisors, and 643 remaining teeth (13.3%) showing bone loss (Table 2). Frequency, percentage, and amount of bone loss were identified more obviously on the proximal surfaces (mesial and distal) than on the buccal and oral surfaces of the first molars in both upper and lower jaw (Table 2). The respective figures were 55 (8.2%), (0.811 mm) and 57 (8.6), (0.85 mm) on the mesial and distal sites compared to 11 (1.6%), (0.327 mm) and 11 (1.6%), (0.264 mm), respectively, on the buccal and lingual sites, as shown in Table 2. The highest overall amount of bone loss was seen on the mesial and distal aspects of the incisors (5.67 mm and 5.59 mm). Meanwhile, the labial and lingual aspects of the mandibular incisors recorded greater respective amounts of bone loss (3.23 mm and1.89 mm) compared to the labial and palatal aspects of the maxillary incisors (0.82 mm and 0.406 mm). In addition, higher respective amounts of bone loss were recorded for distal and mesial alveolar bone loss of maxillary incisors (1.31 mm and 1.28 mm) than for buccal and lingual (0.82 mm and 0.406 mm), while respective bone losses for mesial and distal mandibular incisors (4.36 mm and 4.31 mm) were greater than for labial and lingual incisors (3.23 mm and 1.89 mm). Further, the majority of bone loss sites demonstrated a horizontal pattern of bone loss around first molars and incisors rather than a vertical pattern, at a level that was significant (*p* = 0.00), (Table 2).

Figure 2A,B demonstrate the mean amount of bone loss in mm at each site around the maxillary and mandibular first molars and incisors. The highest amount of bone loss was detected around mandibular incisors, followed by the maxillary incisors, while the least amount was seen around the first molars. In addition, higher amounts of bone loss were observed at the proximal surfaces of all the studied teeth than at the facial and lingual/palatal aspects.

The CBCT images scanned in this study showed low rates of furcation involvement at the maxillary and mandibular first molars. The bilateral site distribution in terms of frequency, percentage, and amount of alveolar bone loss around each two corresponding bilateral teeth is presented in Table 3, which shows significant differences between the four sites of the teeth scanned in this study (Pv < 0.05).

The Mann–Whitney (two-tailed) test showed a statistically significant difference (*p* < 0.05) in the percentage of bone loss between females and males around all first molars and incisors, except for tooth number 12, where a non-significant difference (*p* = 0.31) in the rate of bone loss between males and females was recorded (Figure 3).

The outcome data were divided into six age groups (Table 1), and the significantly different values for the amount of bone loss between the groups are presented below in Table 4. For example, in the analysis of the outcome data, tooth n46 showed non-significant differences between the age groups, while the only significant values were found between group (1) and group (6) and between group (2) and (6), with *p*-values of (*p* = 0.02 and 0.01), respectively. Any other differences between groups were non-significant, and therefore, they were not included in the table.

## 4. Discussion

A molar–incisor pattern of periodontitis of periodontitis can cause permanent and rapid destruction of the periodontal tissue [2]. The incisor region, and in particular, the maxillary anterior region, is of growing major concern due to its esthetic relevance [16]. Therefore, it is necessary to devise an early method of detection in order to stop progression of the disease and maintain the periodontium in a state of health later in life. Furthermore, the consensus report of the 2017 Classification World Workshop considered the amount of alveolar bone loss as a direct sign of the severity and progression of periodontal destruction [17].

In the present study, 250 valid CBCT images were scanned, with (*n* = 2578) teeth and (*n* = 10,312) sites interpreted to explore some more detailed radiographic characteristics of the alveolar crest area. In the current study, there was a discrepancy in sample size between females, *n* = 153 (61.2%) and males, *n* = 97 (38.8%) and between the respective numbers of first molars and incisors, *n* = 668 (25.9%) compared to *n* = 1910 (74.1%); thus, there were three-times as many first molars in the female group as in the male group.

Meanwhile, the number of extracted posterior teeth, in particular first molars [18], exceeded the number of extracted incisors in this study, which could probably be due to the swift extraction of problematic posterior teeth because of their lack of visibility compared to the incisors, which are of esthetic importance. Patients mostly prefer extraction of invisible teeth when they are suffering from severe pain of pulpitis and if they cannot afford the high cost of highly advanced therapy in private dental clinics. In particular, posterior root canal therapy is not available in public health centers, and the private sector cost is higher than most patients’ ability to pay.

This could explain why the number of missing first molars was much higher than that of incisors in both sexes. Furthermore, the number of missing first molars was higher in females than males, and the number of missing incisors was lower in females than males, which reflects females’ greater concern about esthetic issues.

Numerous studies have examined alveolar bone loss and its features at different sites and in different ways; some have addressed the furcation areas, whereas others have examined only the proximal defects [19,20] or labial and lingual defects around the teeth. Published articles have shown an increasing rate of tooth loss in elderly males [21]. Therefore, it can be accepted that some missing first molars were extracted as a result of attachment loss and alveolar bone loss due to periodontal disease, especially among patients of low economic status and limited awareness of post-extraction complications [22,23].

In the current study, the highest degree of bone loss was observed at the proximal surfaces of all examined teeth compared to B/L and P/L aspects, which could be due to the inaccessibility of the proximal areas to plaque control methods, while higher rates of proximal caries and overhanging restoration margins at the interproximal areas could be considered additional retentive factors for proximal bone loss [24]. However, M.A. Alsaegh and A.W. Albadranii (2020) reported that periodontal disease was the main cause of the extraction of mandibular incisor teeth, and this could be due to anatomical factors related to the morphology of soft and hard tissues of the mandibular anterior region around the incisor teeth [21]. In this study, the labial aspects of the mandibular central incisors showed high amounts and percentages of bone loss compared to the labial aspects of the maxillary incisors. This could be attributed to the morphologic site differences in the alveolar process and thin cortical bone on the labial surfaces of the mandibular incisors, as well as a higher incidence of fenestration in the lower anterior teeth compared to the upper anterior teeth [25,26].

In this study, the rate of the furcation bone loss scanned at first molar teeth was limited as a large number of the first molars among the total sample had been extracted; however, no significant differences were observed in furcation involvement between maxillary and mandibular first molars (Pv = 0.300) or between left and right side. Whereas an epidemiologic study of a Swedish adult population by Najim et al. (2016) reported the highest prevalence of furcation involvement at the maxillary first molars [27], the current study found the highest prevalence of furcation bone loss among the first molars to be at the mandibular left first molars (2.8%). Furthermore, Wenjian et al. (2018) used axial CBCT reconstructions of existing CBCT scans of eighty-three patients with chronic periodontitis, along with intraoral (periapical and/or bitewing) and clinical examination, to evaluate furcation involvement on buccal and palatal/lingual sites. CBCT at axial access provided more accurate assessment, with bone loss measurement up to two decimals in millimeters [15].

Alveolar bone defects in periodontitis have attracted more attention in the new classification [28]. In this study, the majority of sites that displayed bony defects around first molars and incisors were horizontal rather than vertical, and the difference was statistically significant (*p* = 0.00), at (*n* = 37 vs. *n* = 19) and (*n* = 434 vs. *n* = 56) for molars and incisors, respectively. This could be explained by the slow progression of the disease and bone loss process [19,29]. The majority of horizontal bone loss was seen on mandibular incisors, which had (*n* = 320 vs. *n* = 37) vertical defects, with a total of (*n* = 213) sites on the labial aspects.

Generally, the rate of bone loss around the individual teeth scanned was significantly higher in males than females in the current study, which agrees with several reports on the rate of alveolar bone loss in periodontitis [25,26,30,31]. However, this result contradicts that of a study performed on a middle-aged (40–59 years) group of Chinese patients with chronic periodontitis, assessed using CBCT images [32].

Papapanou and coworkers (2008), who studied the mean annual rate of bone loss among 200 periodontally involved patients over ten years and according to their age [33], reported a significantly higher rate of bone loss among the elderly than young people. Similarly, our sample showed an increasing rate of bone loss with age. The literature has consistently identified a higher prevalence of alveolar bone loss with aging [34,35], especially in women with osteoporosis [25].

It is suggested that further similar studies need to be conducted using a larger sample size and multicenter data to overcome the discrepancy in the numbers of females vs. males and numbers of first molars vs. incisors, which may have led to discrepancies in the results of the study. Furthermore, it is suggested that future studies should be conducted using clinical data to support the radiographic scanning and to provide an additional method of evaluation of the complex architecture of the area and assist achieving the appropriate diagnosis and treatment planning.

## 5. Conclusions

The limitations of this study, such as the high number of posterior teeth extractions, which led to an unequal distribution of the data among the whole sample, in particular in terms of discrepancies in sample size between males and females and between first molar incisors, were discussed in the Discussion Section. This may support the idea of conducting further multicenter studies on prevalence, rate, and pattern of bone loss at first molar and incisor regions with a greater sample size. However, scanning of the 250 clearly defined, full upper and lower arch CBCT images randomly selected for this study indicated that the rate and amount of bone loss were vertical rather than horizontal in pattern and that vertical bone loss was focused mainly in the interproximal areas of the first molars. However, the highest recorded amounts of bone loss were at the proximal and labial aspects of the mandibular incisors. Furthermore, younger age groups displayed significantly higher rates and amounts of bone loss than older groups, with a slight predilection for males.

## Figures and Tables

**Figure 1 diagnostics-12-01536-f001:**
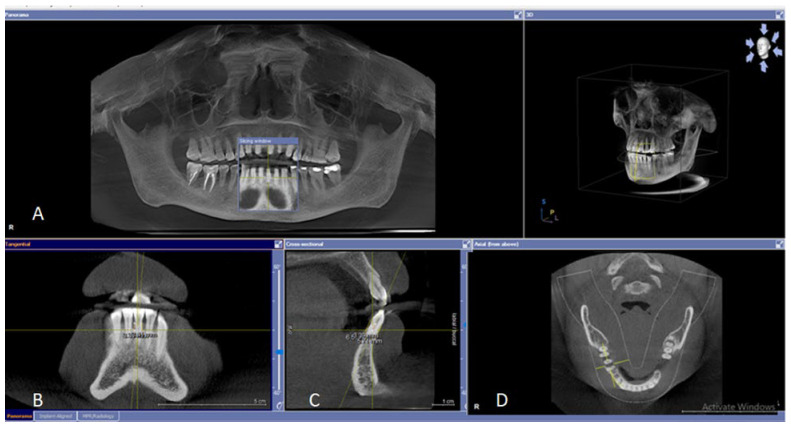
Scanning dentogingival region by CBCT to detect the osseous lesions and measure the amount and pattern of osseous defect. (**A**) An overview of the region by Orthopantomography (OPG) to determine the level of alveolar crest, in particular at the proximal and furcation region. (**B**) Tangential view for inspecting and measuring interproximal osseous defects. (**C**) Cross-sectional view for the detection of osseous defects in buccal and lingual aspects. (**D**) Axial view for the detection of osseous defects at furcation areas.

**Figure 2 diagnostics-12-01536-f002:**
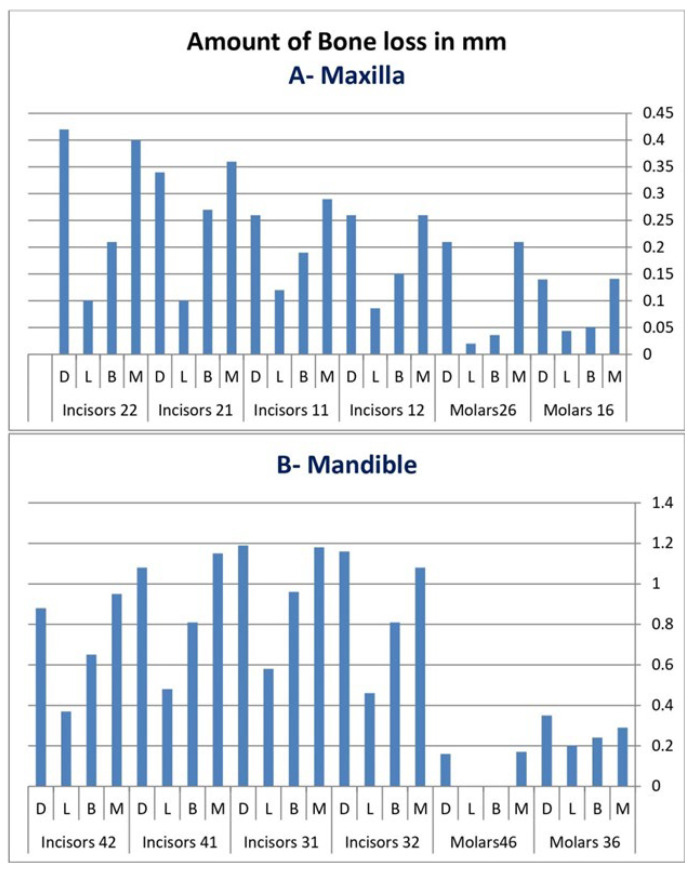
Amount of bone loss in mm at each individual aspect around the maxillary (**A**) and mandibular (**B**) 1st molars and incisors.

**Figure 3 diagnostics-12-01536-f003:**
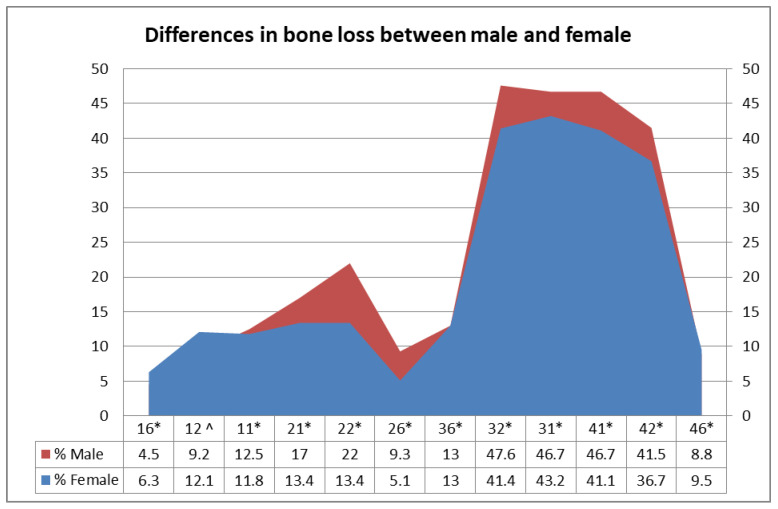
Percentages of bone loss among males and females. Mann–Whitney test: (*) indicates highly significant statistical differences; (^) indicates non-significant differences.

**Table 1 diagnostics-12-01536-t001:** Demographic profile of the study sample.

Demographic Profile of the Study Sample				
(*n*) CBCT	250		(*n*) Teeth examined = 6067	
Number of present 1st molars and incisors teeth and sites examined	Total	2578 Teeth 10,312 Sites	Maxilla	1313—T
				5252—S
			Mandible	1265—T
				5060—S
Number of missing 1st molar and incisors T. Total 422	1st Molars	332 78.67%	Male	136 (32.23%)
			Female	196 (46.44%)
	Incisors	90 21.32%	Male	52 (12.32%)
			Female	38 (9.0%)
Number of 1ST molars and incisors present	First molars	668 (25.9%)	Maxilla	370 (14.35%)
			Mandible	298 (11.56%)
	Incisors	1910 (74.1%)	Maxilla	943 (36.57%)
			Mandible	967 (37.515)
Sex	Male		Female	
	97 (38.8%)		153 (61.2%)	
Age	18–79		Mean + SD	40.68 ± 13.12
Group-1	18–20	*n* = 14 (5.6%)		
Group-2	21–30	*n* = 49 (19.6%)		
Group-3	31–40	*n* = 66 (26.4%)		
Group-4	41–50	*n* = 65 (26.0%)		
Group-5	51–60	*n* = 35 (14.0%)		
Group-6	>60	*n* = 21 (8.4%)		

**Table 2 diagnostics-12-01536-t002:** Site distribution, frequency, percentage, and total amount of bone loss in mm revealed around maxillary and mandibular 1st molars and incisors and patterns of bone loss among sites of these two groups of teeth in the upper and lower jaws.

Frequency, Percentage, and Pattern of Bone Loss									
Frequency & % by Teeth		1st Molars = 58 (1%)				Incisors = 540 (8.9%)		Total = 23.2%	
By Sites		1st Molars—total				Incisors—total			
Site		M	B	L	D	M	B	L	D
Frequency		55	11	11	57	538	276	229	539
Percentage		8.2	1.6	1.6	8.6	28.2	14.6	12	28.3
Amount (mm)		0.811	0.327	0.264	0.85	5.67	4.050	2.296	5.59
Pattern total No. of 1st molars *n* = 668 T	H	37				Pattern total No. of incisors *n* = 1910 T	H	434	
	V	19					V	51	
	Non-affected	612			^ Pv 0.00		Non-affected	1370	^ Pv 0.00
Maxilla		Maxillary 1st molars				Maxillary incisors			
Site		M	B	L	D	M	B	L	D
Frequency		23	4	4	24	129	65	55	127
Percentage		6.2	1.1	1.1	7	13.7	7.1	5.8	13.5
Amount (mm)		0.351	0.087	0.064	0.35	1.31	0.82	0.406	1.28
Pattern Maxillary 1st molars *n* = 370	H	17			^ Pv 0.001	Pattern maxillary incisors *n* = 943 T	H	114	^ Pv 0.001
	V	6					V	14	
	Non-affected	347					Non-affected	815	
Mandible		Mandibular 1st molars				Mandibular incisors			
Site		M	B	L	D	M	B	L	D
Frequency		32	7	7	33	409	213	174	412
Percentage		10.7	2.3	2.4	11.1	42.3	21.9	18	42.2
Amount (mm)		0.46	0.24	0.20	0.51	4.36	3.23	1.89	4.31
Pattern maxillary 1st molars *n* = 298 T	H	20			^ Pv 0.001	Pattern mandible incisors *n* = 967 T	H	320	^ Pv 0.001
	V	13					V	37	
	Non-affected	265					Non-affected	610	

^ Kruskal–Wallis test was used to compare the mean values of bone loss and pattern. First molars (horizontal and vertical) recorded highly significant differences.

**Table 3 diagnostics-12-01536-t003:** Comparison of bilateral site distribution of the corresponding maxillary and mandibular teeth for frequency, percentage, and amount of bone loss at the 1st molars and incisors, plus 1st molars’ furcation defects, among the total number of sites (nS) of all teeth (nT) examined in this study.

Site	M		B		D		L/P			Site	M		B		D		L/P		
Teeth Number and Sites	Frq	%	Frq	%	Frq	%	Frq	%	PV	Teeth Number and Sites	Frq	%	Frq	%	Frq	%	Frq	%	PV
11 nT-232 nS-928	29	12.6	11	4.7	28	12.1	14	6	0.01	21 nT-237 nS-948	35	14.8	21	8.4	35	14.8	16	6.8	0.01
Amount mm	0.29		0.19		0.12		0.26			Amount mm	0.36		0.27		0.10		0.34		
12 nT-236 nS-944	25	10.6	11	4.7	25	10.7	10	4.2	0.01	22 nT-238 nS-952	40	16.8	18	7.6	39	16.4	15	6.3	0.01
Amount mm	0.26		0.15		0.09		0.26			Amount mm	0.40		0.21		0.10		0.42		
16 nT-178 Ns-712	11	6.1	2	1.1	10	5.6	2	1.1	0.02	26 nT-192 nS-768	12	6.3	2	1	13	6.8	2	1	0.01
Amount mm	0.141		0.051		0.044		0.14			Amount mm	0.21		0.036		0.02		0.21		
16	Fur *n* %	3 1.2%								26	Fur *n* %	1 0.4%							
31 nT-238 nS-948	107	45	60	25.3	108	45.4	50	21	0.01	41 nT-238 nS-952	103	43.3	54	22.7	103	43.3	41	17.2	0.01
Amount mm	0.18		0.96		0.58		1.19			Amount mm	1.15		0.81		0.48		1.08		
32 nT-247 nS-988	105	42.5	53	21.5	107	43.5	46	18.6	0.01	42 nT-244 nS-988	94	38.5	43	17.6	92	37.7	37	15.2	0.01
Amount mm	1.08		0.81		0.46		1.16			Amount mm	0.95		0.65		0.37		0.88		
36 nT-146 nS-584	18	12.3	7	4.8	19	13.1	7	4.8	0.036	46 nT-152 nS-616	14	9..2	1	0.7	14	9.3	0	0	0.01
Amount mm	0.29		0.24		0.20		0.35			Amount mm	0.16		0.0		0.0		0.16		
36	Fur *n* %	7 2.8%								46	Fur *n* %	1 0.4%			• Friedman’s Test for Furcation.16, 26,36,46				
															Pv = 0.300				

• Friedman’s test on the mean rank of 16, 26, 36, and 46 showed no significant differences in the amount of bone loss at the furcation areas of the 1st molars (Pv = 0.300).

**Table 4 diagnostics-12-01536-t004:** Significant values for the amount of bone loss among the age groups around the 1st molars and incisors—ANOVA test.

Tooth numbers and statistical differences between age groups											
16 (0.001)		12 (0.001)		11 (0.001)		21 (0.001)		22 (0.001)		26 (0.001)	
1*5	0.001	1*5	0.002	1*5	0.05	1*5	0.001	1*5	0.001	1*5	0.001
2*5	0.001	1*6	0.001	1*6	0.001	1*6	0.02	1*6	0.001	2*5	0.001
3*5	0.001	2*5	0.001	2*5	0.01	2*4	0.001	2*4	0.002	3*5	0.001
4*5	0.001	2*6	0.001	2*6	0.001	2*5	0.001	2*5	0.001	4*5	0.001
5*6	0.001	3*5	0.001	3*5	0.001	2*6	0.001	2*6	0.001	5*6	0.001
		3*6	0.001	3*6	0.001	3*4	0.01	3*4	0.003		
		4*6	0.001	4*6	0.001			3*5	0.001		
								3*6	0.001		
36 (0.001)		32 (0.001)		31 (0.001)		41 (0.001)		42 (0.001)		46 (0.084)	
1*5	0.001	1*4	0.001	1*4	0.001	1*4	0.001	1*4	0.007	1*6	0.02
1*6	0.01	1*5	0.001	1*5	0.001	1*5	0.001	1*5	0.001	2*6	0.01
2*5	0.001	1*6	0.001	1*6	0.001	1*6	0.001	1*6	0.001		
2*	0.01	2*4	0.001	2*4	0.001	2*4	0.001	2*4	0.001		
3*5	0.001	2*5	0.001	2*5	0.001	2*5	0.001	2*5	0.001		
3*6	0.01	2*6	0.001	2*6	0.001	2*6	0.001	2*6	0.001		
4*5	0.001	3*4	0.001	3*4	0.001	3*4	0.001	3*4	0.001		
		3*5	0.001	3*5	0.001	3*5	0.001	3*5	0.001		
		3*6	0.001	3*6	0.001	3*6	0.001	3*6	0.001		
		4*5	0.001	4*5	0.001	4*5	0.001	4*5	0.003		
		4*6	0.001	4*6	0.001	4*6	0.001	4*6	0.001		

## Data Availability

Date are available with the author at reasonable request.

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
