# Peer review of "First Molars–Incisors Rate and Pattern of Bone Loss: A Cross-Sectional Analysis of CBCT Images"

_diagnostics, 2022, doi:10.3390/diagnostics12071536_

Round 1

Reviewer 1 Report

1.      The images were obtained from only one private dental center. Multicenter data would be added and more valued.

2.      Only 250 CBCT images were analyzed and divided into six age group. Did the results reach the bottom line of statistic power?

3.      It is not clear what kinds of section or axial view were assessed.

4.      How many investigators performed this survey?

5.      How to resolve the discrepancy of results?

6.      How about the Kappa value of intra- and inter-investigator?

7.      It is not clear the repeat number of this descriptive observation.

8.      The stratified analysis of sex and six age groups should be displayed.

9.      Without the data of clinical examination, diagnosis, and oral habits, the results can not provide the cause-related relationship.

10.  Without the control group, this observation is rather a case series than as a full length article.  

Author Response

Response to Reviewer 1

 Re: diagnostics-1738659 “First molars-incisors rate and pattern of bone loss: a cross sectional analysis of CBCT images”

The author would like to thank the editor and the reviewers for their thoughtful and meaningful suggestions that surely helped me to further improve this manuscript. Based on the reviewers’ suggestions, I have revised the manuscript and fully responded to all the Referee’s comments. I have highlighted the changes to the manuscript and easy to track. Please find a point-by-point response to the reviewers’ comments.

Response to points raised by the reviewer 1:

  1. The images were obtained from only one private dental center. Multicenter data would be added and more valued.

Authors Response:  I am grateful for this thoughtful suggestion. However, I tried my best to include a few dental practice centers and as there are only a few practices that have CBCT in the city and unfortunately not all of them cooperate and are happy to share their data. Of course, this valuable point will be considered in future studies.

  1. Only 250 CBCT images were analyzed and divided into six age groups. Did the results reach the bottom line of statistic power?

Author response: Although a higher sample number provides a better comparison, however, if we divide the sample into 6 groups the result will be more than 40 for each group although some groups include much higher or much lower than (40), as an average so the statistical tests can deal with this despite

  1. It is not clear what kinds of section or axial view were assessed.

Author response: in the current study detection of furcation bone loss is based on the Orthopantomography (OPG) and axial access of CBCT as in a previous study (Reference 15). Clarification added (Line 105-107) also (line 272-277).

  1. How many investigators performed this survey?

Author response: One principal investigator performed the study with the assistance of an oral radiologist and a periodontist. Their help is appreciated in acknowledgment section (Line 320-325).

  1. How to resolve the discrepancy of results?

Author response: Very good point raised by the reviewer. In the discussion section, the discrepancy in the sample size and number of first molars vs incisors was mentioned (line 227-242), I acknowledge that in our result and explanation with future suggestions have been added to the discussion section (line 297-303). Thanks again.

  1. How about the Kappa value of intra- and inter-investigator?

Author response: Thank you very much for your keen observation, as a matter of fact, inter and intra examiner calibration was performed for this study but unfortunately it was forgotten to be added. Currently, this section was added to the methodology section. The inter-examiner calibration was achieved at a level of 85% during a training course with a radiologist, and  Since the data was scanned by one expert examiner only intra investigator was performed at a level of 87% competency (Line 126-132). Thank you very much for reminding me to add this important section to the methodology section of this research.

  1. It is not clear the repeat number of this descriptive observation.

Author response: Some repeated numbers are mostly associated with different presentations of the data and as it is an observational study, I tried to be clearer to the reader. Thanks.  

  1. The stratified analysis of sex and six age groups should be displayed.

Author response: Thank you very much for giving me the opportunity to explain this important point. As a matter of fact, a descriptive presentation of sex is parented in Figure 3 and at the lower bar of the Figure, the statistical analysis is presented. It was performed by applying Mann-Whitney Test: (*) indicates highly significant statistical differences, (^) indicates non-significant differences. As it is clarified in the legend of Figure 3, (Line 214-215). Furthermore, the statistical analysis of the six age groups are also presented in Table 4.  Only the Significant values for the amount of bone loss among the age groups around the 1st molars and incisors – ANOVA test as presenting all values was not possible because of a very large space was required for that, therefore, only the significant values were presented in the table (line 211-212). 

  1. Without the data of clinical examination, diagnosis, and oral habits, the results can not provide the cause-related relationship.

Author response: Thank you for this valuable comment. It is true that clinical data would add further value to the study. However, the aim of this observational retrospective study was to look at the pattern and rate of bone loss. Although clinical data will add more crucial advantages for supporting the radiographic scanning and to provide an additional method of evaluation of the complex architecture of the area and assist in achieving the appropriate diagnosis and treatment planning, Suggestion has been added to the discussion section (line 297-303). Yes, it doesn’t provide a cause-related relationship as the study does not include the cause of bone loss it is just a cross-sectional interpretation of the rate and morphology of bone loss at 1st molars- incisors of randomly selected CBCT images.

  1. Without the control group, this observation is rather a case series than as a full length article.

Author response: This is a cross-sectional study to determine the rate and pattern of bone loss around 1st molars-incisors among selected CBCT images among a selected age group of patients. The numbers of samples examined are far beyond a case series study. Furthermore, I agree that having a control group would be of great interest; however, the nature of cross-sectional study is like that without a control group like in a case-control study.

The author would like to thank the reviewer for the constructive feedback to improve the quality of this manuscript. I believe that the quality of the revised manuscript is improved and it will be acceptable for publication.

Sincerely.

Reviewer 2 Report

manuscript requires extensive language editing, kindly use the help of native English speakers. 

scientific content also to be improved

conclusion should be more of inference and not the results obtained

rest of the specific comments are given in the pdf attached below

Author Response

Re: diagnostics-1738659 “First molars-incisors rate and pattern of bone loss: a cross sectional analysis of CBCT images”

The author would like to thank the editor and the reviewers for their thoughtful and meaningful suggestions that surely helped me to further improve this manuscript. Based on the reviewers’ suggestions, I have revised the manuscript and fully responded to all the Referee’s comments. I have highlighted the changes to the manuscript and easy to track. Please find a point-by-point response to the reviewers’ comments.

Response to points raised by the reviewer 2:

manuscript requires extensive language editing, kindly use the help of native English speakers.

Author’s Response: Thank you very much for your suggestion and the manuscript has been subjected to an extensive proof read proof read process by a native English speaker.

Scientific content also to be improved       

Author response: Thank you for the suggestion and the scientific content of the manuscript was improved by adding new paragraphs to the introduction (Lines 61-67), (Line 68-70) and (76-84), Further clarifications with a reference was added to the methodology section (Line 105-107) and adding inter and intra examiner calibration to the same section (Line 126-132). We also added Sample size calculation to the Materials and Methods section (Line 134-139). Besides more clarifications, adding paragraphs with references, and suggestions for future study were added to discussion section as highlighted in the manuscript (Line 227-242), (Lines 257-264), (Line 272-277) and (Line 297-303). Furthermore, the conclusion section was amended as well (Line 306-318).

Conclusion should be more of inference and not the results obtained

Author response: Thank you very much for your valuable suggestion about the conclusion section. The section is amended according to your instruction no it reveals better insight into the study and its outcome, thanks again, (Line 306-318).

Response to rest of the specific comments are given in the pdf attached by reviewer 2 and here attached below

1-         Full stop replaced with comma (Line 13)

2-         Keywords amended to alphabetic order (Line 18)

3-         The manuscript amended by a native English speaker.

4-         The word (Clear) changed to (clearly defined), (Line 27)

5-         Revised to 18 to 70 years, (Line 27)

6-         Bone loss considered at 2 mm apical to CEJ - changed to beyond 2 mm apical to the CEJ, (Line 29)

7-         It means that the proximal surfaces of the 1st molars revealed higher amount of bone loss than their buccal and Palatal/lingual surface. While the highest amount of bone loss was at the proximal surfaces (mesial and distal) of mandibular incisors, (4.36 mm and 4.31mm) respectively and the labial aspects of the mandibular incisors (3.23 mm) (Line 32-36) as shown and highlighted in Table 2 (Line 177-182).

8-         Conclusion – the highest amount was 4.36 mm mesial and 4.31 distal of the mandibular incisors and Labial of the incisors (3.23mm) as shown and highlighted in Tab 2.

            Conclusion: please note that conclusion has been amended according to your suggestion (Line 37-41) key words also amended alphabetically (Line 42), thanks again

In the introduction section few paragraphs were added with references (Line 61-71), (Line 76-84) and (REF, 8, 9, 10, 11 and 12) as highlighted in the references section and found in the text.

9-         Sample size calculation was done and it is added to the manuscript (Line 135-139)

10-       Age Group is amended to (18 to 70), (Line 90)

11-       Amended as (while furcation defects were simply recorded as presence or absence of bone loss. (Orthopantomography OPG images used for detection of bone loss at the furcation areas, and confirmation was done by CBCT) according to references (15) that used the axial view for detection of furcation bone loss – (A retrospective study on molar furcation assessment via clinical detection, intraoral radiography and cone beam computed tomography. (line 105-107) with reference 15 as highlighted in the text.

            Two paragraphs for Inter and intra examiner calibration and sample size calculation were added to the Material and methods section (Line 126-132) and (Line 134-139)

12-       CBCD amended to CBCT (Line 192)

13-       Thank you for this important note the sentence is changed to (A molar-incisor pattern of periodontitis can cause permanent and rapid destruction of the periodontal tissue. And reference changed to reference 2. (Line 217-218)

14-       Thanks for give me the opportunity to clarify and explain this point. As a matter of fact, it couldn’t be rational, however, in the non-developing countries and in our country could be be rational specially because of high cost of root canal therapy of the posterior teeth in private clinic and while in public centers posterior root canal is not performed, and patients at low economic slandered usually not afford the severe pain of pulpitis meanwhile, when he couldn’t afford the high cost of root canal therapy of the posterior teeth. Besides in this country patients prefer esthetic on function (Line 227-242).

15-       Thanks again for this comment, It sounds yes, female concern more about esthetic than male and to me this is the only explanation for high rate of extraction of posterior teeth in female than in male). A similar explanation was given by Sahibzada HA, 2016 in his study about pattern and causes of tooth extraction in patients reporting to a teaching dental hospital of Islamabad, (Line 231- 238), REF 18 .

16-       Main case of extraction amended to the main cause of extraction (Line 256), Thanks for that.

17-       We usually encounter numerous case of extensive recession at mandibular central incisors due to high frenum attachment and lack of keratinized gingiva at the mandibular anterior region and perhaps combined with bone dehiscence at this area. This is also referred to Alsaegh et al. 2020 – (Reference 21 and REF, 25,26) (Line 257-264)

18-       Conclusion,

 Conclusion section is amended according to your instructions (Line 306-318)

The author would like to thank the reviewer for the constructive feedback to improve the quality of this manuscript. I believe that the quality of the revised manuscript is improved and it will be acceptable for publication.

Sincerely.

Reviewer 3 Report

The authors aimed to determining the prevalence, frequency and pattern of bone loss at first molar-incisor areas using CBCT. In detail, a retrospective, cross-sectional analysis inspected alveolar bone loss at four aspects (mesial, distal, mid-buccal mid-lingual) around all the remaining teeth, highlighted at the molar/incisors regions and compared to the presence of sites around the other teeth. The structure of the manuscript appears adequate. The methodology is well described with enough experimental data and results to support the work.

Referee's suggestions: Please check typos and grammar thorough the text. Please remove redundant sentences within all the text. The Conclusion Section paragraph needs to be improved. Please also add some "take-home message".

Author Response

Re: diagnostics-1738659 “First molars-incisors rate and pattern of bone loss: a cross sectional analysis of CBCT images”

The author would like to thank the editor and the reviewers for their thoughtful and meaningful suggestions that surely helped me to further improve this manuscript. Based on the reviewers’ suggestions, I have revised the manuscript and fully responded to all the Referee’s comments. I have highlighted the changes to the manuscript and easy to track. Please find a point-by-point response to the reviewers’ comments.

Referee's suggestions: Please check typos and grammar thorough the text. Please remove redundant sentences within all the text.

 Author response: The manuscript Language is amended by a native English proofreader. Thanks for your suggestion.

The Conclusion Section paragraph needs to be improved. Please also add some "take-home message".

Author response: Thank you Very much, the conclusion section has been amended (Line 306-318).

The author would like to thank the reviewer for the constructive feedback to improve the quality of this manuscript. I believe that the quality of the revised manuscript is improved and it will be acceptable for publication.

Sincerely.

Round 2

Reviewer 2 Report

still the language has to be improved, look into grammatical errors such as starting a sentence with small letter especially in abstarct.

kindly provide footnotes for table 3,5 and mention it as table 3,5